

# Early Permian longitudinal position of the South China Block from brachiopod paleobiogeography

Robert J. Marks[1], Nicolas Flament[1], Sangmin Lee[1], and Guang R. Shi[1]

[1]Environmental Futures, School of Science, University of Wollongong, New South Wales 2522, Australia.

**Correspondence:** Robert Marks (rm181@uowmail.edu.au) and Nicolas Flament (nflament@uow.edu)

**Abstract.** Knowledge of the past location of tectonic plates is essential to understanding the evolution of climate, ocean systems, and mantle flow. Tectonic reconstructions become increasingly uncertain back in geological time. Paleomagnetic data constrain the past latitude of continental blocks, however, their past longitude is unconstrained. For example, the longitude of the South China Block during the Early Permian is unknown. Paleobiogeographic data, which have long been used in tectonic reconstructions, make it possible to evaluate the faunal similarity between continental blocks. In this study, we use the Early Permian global brachiopod distribution from the Paleobiology Database to evaluate the correlation between faunal similarity and physical distance of continental blocks for three distinct tectonic reconstruction models. We use this approach to assess which of the three tectonic scenarios places the South China Block in a location that best accounts for the Early Permian brachiopod distribution data. Based on this analysis, the preferred tectonic reconstruction places the South China Block in a central position within the Paleo-Tethys Ocean instead of on its outskirts. The framework developed in this study is openly available and our approach could be applied to other tectonic blocks, time periods, and faunal data.

## 1 Introduction

Plate tectonic motions strongly influence Earth's internal and external systems (Valentine and Moores, 1970; Müller et al., 2008; Flament et al., 2017; Dutkiewicz et al., 2024), making it important to develop robust models of plate tectonic configuration to understand how these systems have evolved through deep time. Constraining the latitude of a tectonic plate through time can be achieved directly via paleomagnetic analysis (Torsvik and Van der Voo, 2002; Krivolutskaya et al., 2016), which determines the distance from the north or south pole. Paleolongitude, however, cannot be constrained in absolute terms, and must be determined in relation to other features. This is commonly done by using either geophysical data from oceanic lithosphere (Scotese, 2004; Seton et al., 2012), which before recent times (< 60 Ma) is progressively lost to subduction through time, or by relating orogens and stratigraphy across plate boundaries (Lehmann et al., 2010), which cannot provide information about isolated tectonic plates.

One method to constrain the relative locations of isolated tectonic blocks and for reconstructions of older times, is paleobiogeographic similarity, which compares the similarity of ancient faunal assemblages between regions as a function of physical distance between tectonic blocks. Fossil data can be used to evaluate the past distance between tectonic plates (e.g. Piccoli et al., 1991; Lees et al., 2002) by assuming that the similarity between the faunal assemblages of two plates decreases as the physical





distance between the plates increases. This assumption is based on the idea that continental configuration influences faunal distribution (Shi, 2001a, b; Zaffos et al., 2017). Tectonic plate configuration controls climate (Valentine and Moores, 1972), ocean circulation (Valentine, 1971; Allison and Wells, 2006), and the locations of land and ocean barriers (Valentine and Moores, 1970), which are all factors that affect faunal distribution. Indeed, faunal similarity is expected to be negatively correlated with
distance (Valentine, 1966), because the spread of species to geographically close areas is easier than to geographically distant areas. Statistical measures of the similarity between two faunal assemblages consider binary presence-absence data to assess either overlapping or unique fauna between two regions (Shi, 1993). Here, we present a new method to apply these quantitative measures globally through a case study of the South China Block during the Early Permian.

The South China Block (SCB) was isolated within large ocean basins throughout the Early Permian (from 299-272 Ma),
which causes a lack of geological evidence linking it to other blocks, thus limiting the data available for constraining the longitudinal position of the plate. In this contribution, we consider the Early Permian location of the SCB for three different plate tectonic reconstructions, with two placing it on the boundary of the Panthalassa Ocean and Paleo-Tethys Ocean (Wright et al., 2013; Matthews et al., 2016) and the other placing it much further west, locating it centrally within the Paleo-Tethys Ocean (Young et al., 2019). Paleomagnetic analysis of the Emeishan Large Igneous Province (Emeishan LIP) in the Mid Per-
mian constrains the paleolatitude of the SCB to low latitudes around the equator (Krivolutskaya et al., 2016) at approximately 260 Ma (Zhong et al., 2014). While this is not a direct latitudinal placement for the Early Permian, it does provide a known latitude immediately following the Early Permian; consequently, the SCB can be assumed to be near this latitude during the Early Permian.

The differences in the proposed position of the SCB among the three reconstructions are largely due to differing methods to
infer its Early Permian longitude. The eruption of the Emeishan LIP is attributed to a mantle plume, which has been suggested to originate from the edges of the African and Pacific Large Low Shear-Wave Velocity Provinces (LLSVPs) (Torsvik et al., 2008). The link between mantle plumes and LLSVPs has been used in some reconstructions to infer the longitude of plates on which Large Igneous Provinces are preserved, by placing such plates at the margin of one of the LLSVPs (Torsvik et al., 2010). In contrast, in some other reconstructions (e.g. Scotese, 2004; Young et al., 2019), the longitude is based on interpretations of
tectonic activity preserved in the rock record. Here we use statistical measures of faunal similarity to assess the compatibility of each of these SCB paleolongitude scenarios with global plate tectonic configuration.

## 2 Data

### 2.1 Plate Tectonic Reconstructions

We consider three tectonic reconstructions that are based on different approaches and present large variations in the location of
the SCB during the Early Permian (Wright et al. (2013); Matthews et al. (2016); Young et al. (2019) labelled here as W13, M16 and Y19 respectively). Reconstruction W13 primarily builds upon previous plate models from Scotese (2004), Golonka (2007) and Seton et al. (2012) by developing an updated location for the Australian plate based on paleoenvironments interpreted from faunal data. The placement of the SCB was an estimated transitional position based on information of tectonic activity



in the rock record (Scotese, 2004). The relatively eastern placement of the SCB in M16 (133ºE at 277 Ma) was based on work by Domeier and Torsvik (2014) which attributes the Emeishan LIP to the western margin of the Pacific LLSVP under the assumption that the LLSVP has remained spatially fixed since the Emeishan LIP emplacement (Conrad et al., 2013). Reconstruction Y19 builds on reconstruction M16, using the same set of tectonic plate geometries, which makes a direct comparison between these two reconstructions possible. The global plate velocities required by the plate tectonic configuration in M16 were deemed to be unreasonably high, with the SCB moving at speeds of up to 40 cm/yr between 260 Ma and 250 Ma, which is tectonically unreasonable (Young et al., 2019; Zahirovic et al., 2015). Reconstruction Y19 uses records of tectonic activity to constrain the SCB longitude to a transitional position (similar to the approach of W13), to define a position which gives more tectonically reasonable plate velocities (Young et al., 2019). All of the three reconstructions use paleomagnetic data as a latitudinal constraint, which is particularly effective because the north-south uncertainty that is typically a concern for paleomagnetic data is less pronounced for these SCB positions as it was situated near the equator. This study provides a chance to test the validity of the different reconstruction models, particularly between M16 and Y19, which tests competing ideas on the mobility of LLSVPs.

We use 277 Ma as the representative age of the Early Permian because W13 reconstructs the Early Permian plate tectonic configuration using paleoenvironments based on 277 Ma. This is due to W13 being based on the works of Golonka et al. (2006) and Golonka (2007), which considers the Early Permian to be 285 - 269 Ma, making 277 Ma the midpoint, with the earliest Permian ages being grouped instead with the latest Carboniferous as many major continental collisions which formed Laurasia and Gondwana began during the Carboniferous and reached maturity during the earliest Permian (Golonka et al., 2006). This causes many of the species present in earliest Permian brachiopod assemblages to be persistent from the Late Carboniferous (Shen et al., 2019), with major changes in South China brachiopod taxonomic composition following sea-level rise during the middle of the Early Permian (Shen et al., 2019). Using this time allows for a better comparison between W13 and the other two reconstructions. At 277 Ma, the SCB was placed further west in Y19 by approximately 3,300 km compared to M16 and 2,500 km compared to W13; the centroid of the SCB is located at longitude 103°E in Y19, 133°E in M16 and 125°E in W13. Because these reconstructed locations are different, we anticipate that their respective compatibility with brachiopod distribution can be established quantitatively.



**Figure 1.** Reconstructions at 277 Ma with distance ranges as concentric circles at 4,000 km, 6,000 km, 8,000 km, 10,000 km, and 12,000 km from the South China Block (SCB). (a) Reconstruction of Young et al. (2019, Y19), (b) Reconstruction of Matthews et al. (2016, M16), and (c) Reconstruction of Wright et al. (2013, W13). Fossil occurrences are shown as green disks, plate outlines as open cyan polygons, continents as filled light grey polygons, and oceans in dark grey. The name of tectonic blocks are abbreviated as follows: Am: Amuria, Au: Australia, Ba: Baltica, CT: Cimmerian Terranes, IC: Indochina, KL-AS: Kunlun - Ala Shan, NAC: North America Craton, NC: North China, SAC: South America Craton, Sb: Siberia, SCB: South China Block, Ta: Tarim.





## 2.2 Brachiopod fossil data from the Paleobiology Database

The distribution of Early Permian brachiopods can be summarised by into three realms with distinct brachiopod faunal as-
semblages (Waterhouse and Bonham-Carter, 1975): Boreal (northern latitudes), Paleo-equatorial, and Gondwanan (southern
latitudes) realms. These realms exhibit biogeographic patterns that match those expected from their continental configuration.
The Gondwanan and Boreal realms are dominated by large continental landmasses and cold-water environments which produce
the expected low diversity, while the Paleo-equatorial realm, which has warm waters and hosts many small island regions, has
a correspondingly high degree of biodiversity (Shen et al., 2013). The consistency between expected biogeographic patterns
based on continental configuration and the biogeographic realms suggests Early Permian brachiopod distribution should match
the expected negative trend of decreasing faunal similarity with distance, making them an appropriate dataset for this study.

Early Permian brachiopod biogeographic data were downloaded from the Paleobiology Database (Paleobiology Database)
on 29/07/2021. Data on "occurrences" were retrieved for the taxa "brachiopoda" with a taxonomic resolution of "lump by
genus" using the following parameters: Interval or Ma range = 299-272, age rule = major, outputs = coordinates, location,
paleolocation. The paleobiogeographic data set consisted of 15,670 fossil occurrences made up of 591 genera, with columns
used for analysis: "accepted_name" (genus name), "lat" (latitude), and "lng" (longitude).

The fossil record is known to be incomplete, although the degree to which this impacts studies using fossil data is contested
(Benton et al., 2000). Preservation biases lead to differential preservation of biota between different environments so that
regions in which large amounts of sedimentary rocks are deposited are more favourable for fossil preservation. Regions pre-
serving Early Permian sedimentary rocks may preserve greater proportions of the regional biota because marine sedimentary
rock volume is positively correlated with biodiversity (Raup, 1976; Crampton et al., 2003). Observed variations in biodiversity
have also been linked to the number of people working on sampling for a region (Sheehan, 1977), with a larger number of
workers in a given region leading to more discovery of specimens. This causes larger biota sets to be known for developed
regions is clear from Figure 1 in which the greatest point densities occur in present-day North America, Europe, China, and
Australia. Such biases are likely to cause apparent differences in biodiversity between regions that are not representative of
true faunal assemblages.

## 3 Methods

### 3.1 Biogeographic indexes

Several binary similarity coefficients have been independently developed (Jaccard, 1907; Simpson, 1960; Lees et al., 2002)
using varying conceptual bases to measure the similarity between two sets of fauna (Hohn, 2018). These coefficients are
useful tools in paleobiogeography to quantify faunal similarity between biogeographic regions by comparing regional biota
datasets based on presence-absence data (Schmachtenberg, 2008; Fallaw and Dromgoole, 1980; Shi, 1993). The underlying
conceptual basis for each index introduces unique statistical biases for each measure, so multiple faunal similarity coefficients
are commonly used in conjunction (Hohn, 2018; Simpson, 1960; Schmachtenberg, 2008).





The Jaccard Coefficient ($JC$, Jaccard, 1907) measures the true similarity between any two sets of fauna assuming that any differences in biodiversity are real. This means that the sampled distribution is assumed to be an accurate representation of the true distribution. It is calculated by dividing the intersection of the biota sets by the union of the biota sets:

$$JC = \frac{S1 \cap S2}{S1 \cup S2} \tag{1}$$

where S1 is the taxa set for the less diverse region (less genera) and S2 is the taxa set for the more diverse region (more genera).

The Simpson Coefficient ($SC$, Simpson, 1960) takes into account that some differences in biodiversity between regions could be a result of sampling bias. This is done by not including the number of genera in the more diverse region in the calculation:

$$SC = \frac{S1 \cap S2}{n1} \tag{2}$$

where $n_1$ is total number of genera in the less diverse region $S_1$.

The Mean Endemism index ($ME$, Lees et al., 2002) accounts for differences in area-biogeographic effects between two regions by averaging the proportions of endemic fauna in each region:

$$ME = \frac{\left(\frac{S1-S2}{n1}\right) + \left(\frac{S2-S1}{n2}\right)}{2} \tag{3}$$

where $n_2$ is the number of genera in the more diverse region $S_2$. $JC$ and $SC$ measure faunal similarity and are expected to be negatively correlated with distance, while $ME$ measures faunal difference and is expected to be positively correlated with distance. For consistency, we use $cME = 1 - ME$ that is expected to be negatively correlated with distance. Values for each of the three indexes $JC$, $SC$ and $cME$ range between 0, indicating complete dissimilarity, and 1, indicating complete similarity.

We considered the natural logarithmic transformation of each index value ($\ln(JC)$, $\ln(SC)$, and $\ln(cME)$) to account for the potential of an exponential relationship between distance and faunal similarity as suggested by Piccoli et al. (1991). The logarithmic indexes have maximum value $\ln(1) = 0$, indicating total similarity, and get progressively more negative as the logarithm argument (linear index value) approaches 0 because $\lim_{x \to 0^+} ln(x) = -\infty$.

To calculate Early Permian faunal similarity indexes, the present-day fossil locations must first be transformed to their Early Permian paleo-locations. The present-day latitude and longitude of each fossil occurrence were used to define a point on the Earth which was then converted to a feature collection saved in a GPlates Markup Language (.gpml) file (Müller et al., 2018) using pyGPlates (Williams et al., 2017). A "point-in-polygon" test was carried out with PyGPlates to assess which tectonic block each point belonged to, and to attribute the corresponding plate ID specific to each tectonic reconstruction. This made it possible to split the data into subsets of biota for each plate and to reconstruct the points back in time using PyGPlates. As a result of the reconstruction of paleo-locations, one fossil occurrence point was located on a W13 plate boundary, and two



points for M16 and Y19 were located within a plate void. These points were discarded as they represented minor portions of the total data set and including them would have introduced errors.

## 3.2 Calculating distance between plates

After each plate is assigned a faunal assemblage we calculate physical distances between each plate to establish the relationship between faunal similarity and distance. The physical distance between plates was calculated as great circle distances along the
surface of a sphere with a constant radius of 6,371 km between plate centroids using pyGPlates. Some plate IDs were assigned to multiple polygons that collectively represented a larger plate. In these cases, a new polygon was created from the multiple plate centroids and the centroid of that new polygon was used as reference for that plate.

A distance limit $d_l$ from the SCB was imposed to obtain more meaningful results. The distance limit must include a large enough sample of plates to obtain a statistically meaningful relationship between distance and similarity (Fig. 2a), however,
larger distances introduce continental land barriers between plates (Fig. 1), which complicates the relationship between biogeographic similarity and distance by making migratory distances much longer than great circle arc distances, because brachiopods are marine organisms. It is also important that plates within a distance limit have, on average, a large enough faunal assemblage to make meaningful comparison with the SCB faunal assemblage (Fig. 2b). We considered distance limits surrounding the SCB from 4,000 km to 12,000 km in 2,000 km intervals (see concentric circles in Fig. 1). A plate was considered within range if the
minimum distance between its centroid and the centroid of the SCB was smaller than the distance limit. If a plate was within range, all fossil occurrences on the plate were considered.

## 3.3 Statistical Tests and Data Processing

All three biogeographic index values and their natural logarithms were calculated for each plate pair that consisted of the SCB and another plate. The Pearson correlation coefficient ($r$ value) was used to determine the strength and direction of
the linear relationship between faunal similarity as measured by the biogeographic indices and physical distance. A negative $r$ value is expected if faunal similarity decreases as physical distance increases. We carried out a one-tailed test to determine statistical significance of the relationship between faunal similarity and physical distance for each distance limit in each tectonic reconstruction. We used the null hypothesis that faunal distribution was random and unrelated to physical distance between plates, which was rejected at the 95% confidence interval if the $p$ value was smaller than 0.05. A one-tailed test was chosen
as it allowed us to test for statistical significance only if the relationship was negative (decreasing faunal similarity as physical distance increases) as we consider a positive global relationship (faunal similarity increases with physical distance) is more indicative of an incorrect plate tectonic configuration rather than a true relationship. The strength of the correlation between faunal similarity and physical distance and the statistical significance of those relationships were then used to determine the most appropriate reconstruction and ideal distance limit within that reconstruction.
Data were prepared and analysed using Python Jupyter Notebooks (Kluyver et al., 2016). The pandas (McKinney, 2010), SciPy (Virtanen et al., 2020) and NumPy (Harris et al., 2020) libraries were used for statistical methods, the pyGPlates (Müller





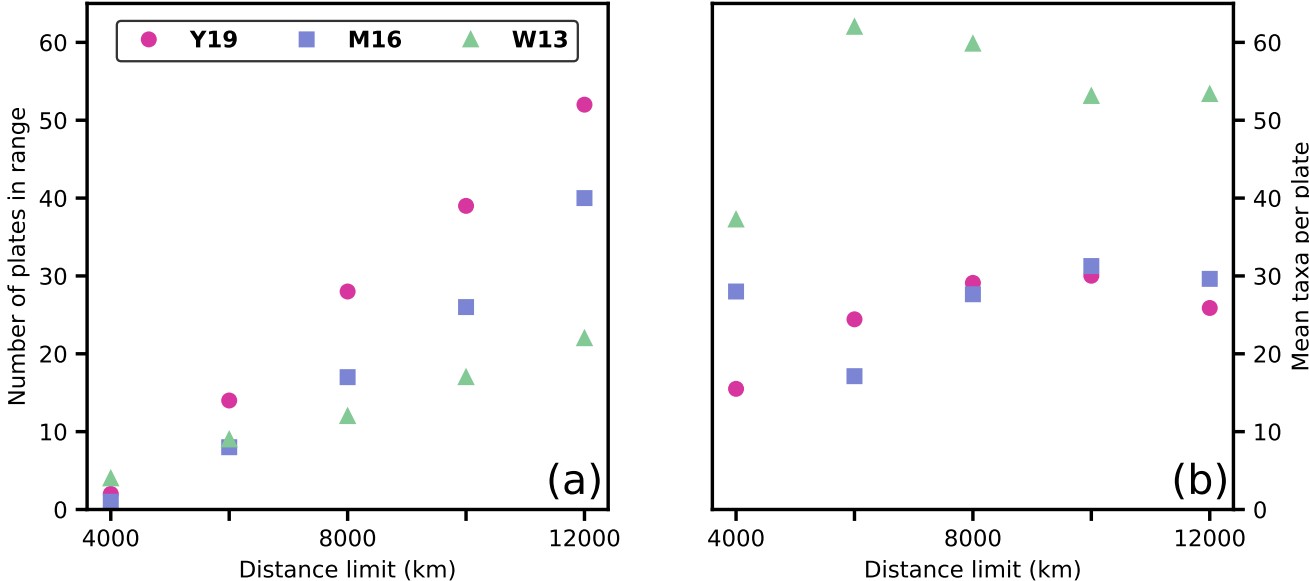

**Figure 2.** Evaluating the effect of the distance limit. (a) Number of plates with centroids within the distance limit from the SCB centroid for each of the three considered tectonic reconstructions W13, M16 and Y19. (b) Average number of genera per plate for the set of plates within a given distance limit for each of the three considered tectonic reconstructions.

et al., 2018; Williams et al., 2017) library for reconstruction and geospatial operations, and the Matplotlib library (Hunter, 2007) to plot results.

# 4 Results

Comparing faunal similarity indexes between the SCB and all other plates in reconstruction W13 reveals a clear inverse relationship between physical distance and faunal similarity (Fig.3), with five of six indices having $r < -0.2$. The relationship is particularly strong for cME (Fig.3c), for which the relationship is statistically significant. These relationships include distant plates for which the migratory distance for brachiopods is much larger than the measured distance, meaning the results can be improved by imposing distance limits on the analysis.

The correlations between faunal similarity and physical distance are expected to be negative for all distance limits, with more negative correlations showing a stronger inverse relationship (white section of Fig. 4), while positive correlations indicate the unexpected relationship of increasing faunal similarity as distance increases (grey section of Fig. 4). Indexes $SC$ and $cME$ and their logarithmic transformations consistently produce stronger correlations than indexes $JC$ and $ln(JC)$. For indexes $SC$ and $cME$, reconstructions W13 and Y19 both produce stronger negative correlations than the M16 reconstruction.

**Figure 3.** All considered biogeographic similarity indexes (and their logarithmic transformations) between the SCB and all 31 other plates in reconstruction W13. $r$ and $p$ values are presented for each index.



**Figure 4.** Pearson $r$ value for each considered index and its logarithmic transformation measuring the anti-correlation of biogeographic similarity and distance when comparing the SCB to all other plates with centroids located within the distance limits on the $x$-axis. Grey region indicates positive correlation, meaning faunal similarity increases with physical distance, which is an unexpected trend.





**Figure 5.** Statistical significance of the anti-correlation between each considered index and its logarithmic transformation and distance from the SCB for all three reconstructions. A relationship is statistically significant ('Yes' in the graph) if the null hypothesis that faunal distribution is unrelated to physical distance between plates is rejected because the $p$ value is smaller than 0.05.



Faunal similarity indexes $SC$, $cME$ and their logarithmic transformations are significantly anti-correlated with distance from the SCB (Fig. 4) in reconstruction Y19 when the distance limit is equal to 8,000 km or 10,000 km (Fig. 5). Other statistically significant anti-correlations are obtained for $JC$ and $cME$ with $d_l$ =12,000 km in reconstruction Y19, and for $SC$ with $d_l$ =6,000 km and $cME$ with $d_l$ =10,000 km in reconstruction W13. These results suggest that the Early Permian longitude of the SCB in reconstruction Y19 is most consistent with the brachiopod biogeographic data among the three reconstructions.

A strong anti-correlation between faunal similarity and physical distance from the SCB was obtained for reconstruction Y19 with $d_l$ =8,000 km (Fig. 4). In detail, the base indexes (Fig. 6a-c) tend to show slightly better correlations than their logarithmic transformations (Fig. 6d-f). The strongest correlation are obtained for indexes $SC$ and $cME$ (Fig. 6b-c), and both of these correlations are statistically significant (Fig. 5b-c). Strong (Fig. 6e-f) and statistically significant (Fig. 5e-f) anti-correlations were also obtained for the logarithmic transformations of $SC$ and $cME$. These results show that there is a statistically significant decrease in faunal similarity as distance increases from the SCB as it is positioned in reconstruction

Y19. In contrast, statistical significance cannot be established for index $JC$ and its logarithmic transformation (Figs 5a, d, 6a) This is likely due to the SCB being a much larger faunal data set than most other plates, which overwhelms the JC index as it considers the union of the faunal data sets (Eq. 1). This suggests that accounting for biases in the number of fossil occurrences on each tectonic block is important to infer significant relationships between the faunal similarity and reconstructed distance

between tectonic blocks because the SC and cME indices both were developed to account for this and have shown more success here.

## 5   Discussion

### 5.1   Position of South China Block in the Early Permian

Of the three considered reconstructions, the Early Permian location of the SCB in reconstruction Y19 is most supported

by the brachiopod fossil records. This is indicated by strong, statistically significant negative correlations between faunal similarity and distance from the SCB for this tectonic reconstruction (Figs 4 and 5). The Early Permian location of the SCB in reconstruction Y19 effectively splits the Paleo-Tethys Ocean into two arms (Fig. 1a), in contrast to its position in the eastern margin of the ocean in reconstructions W13 and M16 (Fig. 1b, c).

Differences in the Early Permian locations of tectonic blocks isolated within ocean basins (e.g. North China, Amuria)

between the considered tectonic reconstructions introduce some uncertainty in the results. As explained in more details below, the main difference between reconstructions M16 and Y19 for the Early Permian is the location of the SCB (Fig. 1). Differences in the reconstructed location of other continental blocks affect the distances to the SCB and the relationships with biogeographic indexes. As such, the comparison between reconstruction W13 and reconstructions M16 and Y19 is less direct.

There are few differences in the geometry of Pangea between reconstructions M16 and Y19 (Fig. 1a, b), with the Siberian

and Baltica blocks slightly further south-east in reconstruction Y19 compared to their locations in reconstruction M16. There are also differences in the location of smaller, isolated blocks. For example, Indochina and Amuria are reconstructed at slightly different distances from the SCB in reconstructions M16 and Y19. Although the North China Block is at similar distances



**Figure 6.** Relationships between biogeographical indexes and distance from the SCB for all 28 plates within $d_l =$8,000 km of the SCB in reconstruction Y19. Biogeographic indexes are shown in the top panels, and their logarithmic transformations are shown in the bottom panels.



from the SCB in reconstructions M16 and Y19, it is to the east of the SCB in reconstruction M16 and to the west of the SCB in reconstruction Y19. In the absence of other significant differences, the major cause for variation in the relationship between
biogeographic indices and distance for reconstructions M16 and Y19 is expected to be the difference in distance of the SCB from Pangean circum-Tethyan blocks between the two reconstructions. Indeed, the SCB is roughly 4,000 km further from the east Pangean coastline (comprised of plates such as Baltica, Armorica, the Pontides, Arabia, and Lut), 2,000 km further from the Cimmerian terranes, and 1,000 km closer to the Australian continental block in reconstruction M16 than in reconstruction Y19 (Fig. 1a, b).

The SCB is reconstructed slightly further to the south-west in reconstruction W13 than in reconstruction M16 and it is reconstructed significantly further to the south-east of its location in reconstruction Y19 (Fig. 1). Even though the locations of the SCB are closer between reconstructions W13 and M16 than between reconstructions W13 and Y19, the correlation results are more similar between reconstructions W13 and Y19 than between reconstructions W13 and M16 (Fig. 4). This result is likely due to other major differences in global paleogeography between reconstruction W13 and reconstructions M16 and Y19
(Fig. 1). The Cimmerian terranes are separated from Pangea and further to the north-east in reconstruction W13, which changes their distribution from the SCB from almost equidistant (the Cimmerian terranes are about 6,000 km away from the SCB in Y19 and 8,000 km away from the SCB in reconstructions M16, Fig. 1) to mostly radially distributed from about 500 km to about 8,000 km away from the SCB in reconstruction W13. The northern Pangean blocks (Siberia and Baltica) are further to the south-east and North China and Amuria are further to the south (such that North China borders the SCB) in reconstruction
W13 than in reconstructions M16 and Y19. These differences may cause the distance to the SCB for many plates to be more similar between reconstructions W13 and Y19 than between reconstructions W13 and M16 (Fig. 4). The different relationships between biogeographic indexes and distance for reconstructions W13 and Y19 (Figs 4 and 5) are due to different distances between the SCB and other plates, including the Cimmerian terranes (see above), the North China Block that is about 4,000 km away from the SCB in reconstruction Y19 but adjacent to it in reconstruction W13, Baltica that is 5,000 km away from the
SCB in reconstruction Y19 and 7,000 km away in reconstruction W13, and the Australian block, that is 4,000 km from the SCB in reconstruction W13 and 6,000 km from the SCB in reconstruction Y19.

The observed decrease in correlation for the $SC$ and $cME$ as the distance range $d_l$ increases for reconstructions Y19 and W13 (Fig. 4) is expected. Indeed, correlation is expected to decrease as sample size increases (which is the case when more plates that carry more fossil occurrences are included). The absence of a trend between correlation and distance limit for
reconstruction M16 could reflect the more complex relationship between sample size and distance limit for that reconstruction (Figs 1b and 2).

Statistical significance could generally be established for greater distance limits (Fig. 5) as increasing the distance limit increases the sample size, which increases the power of the statistical test to identify a relationship (VanVoorhis et al., 2007).



## 5.2 Limitations and Possible Future Improvements

### 5.2.1 Faunal provinciality

Environmental variation across continental blocks can lead to multiple distinct biogeographic regions on a single block. The methods used in this study may lead to inaccuracies when biogeographic regions are selected that are far beyond the distance limit or when multiple regions that should be distinct become combined as one coherent region. This problem is exacerbated in reconstruction W13 with a smaller number of plates and a larger area covered by each plate. Alternate methods to measure

distance and define biogeographic regions may be implemented in future work to improve the method used in this study. Rather than selecting plates within a distance range, distances between each fossil occurrence and the SCB could be measured with only occurrences within the distance limit being selected. This may separate distinct biogeographic regions on large plates, but it also risks splitting blocks (e.g., the Cimmerian terranes in M16, Fig. 1b). Multivariate statistical methods, such as ordination and cluster analysis (Shi, 1993), are other interesting possibilities to address this issue in future improvements

of the framework introduced in this study. These methods may allow for greater resolution in defining biogeographic regions by identifying locations that should be distinct regions rather than defining each region as the entire tectonic block. It has also been found that a distance of 1400 km may be required between islands to develop a distinct faunal province (Shi, 1996). By taking these into account we could potentially improve results for both large plates with multiple faunal provinces and clusters of islands comprised of individual tectonic blocks that may not be distinct faunal provinces.

The break-up of supercontinent Pangea and the demise of Late Paleozoic glaciation initiated in the Early Permian, generating dynamic climatic episodes during the time interval. The episodes have been interpreted to cause increases in biodiversity and changes in faunal assemblages (Zaffos et al., 2017; Angiolini et al., 2005). These biotic changes may be reflected in the data set considered in this study as it covers a long time period ($> 27$ Myr) which would lead to the grouping of multiple faunal assemblages that existed in the same faunal province but at distinct time periods with different climates. While all

three reconstructions have differing continental configurations, most variation is longitudinal differences with corresponding continental blocks having similar latitudes across each reconstruction (Fig. 1). This keeps continental blocks in the same biogeographic realms across each reconstruction as they are defined latitudinally (Waterhouse and Bonham-Carter, 1975; Shen et al., 2013), meaning that the reconstructions should all be affected equally by this concern. This problem could also be limited in the future by selecting periods during which the climate was relatively stable although this could also reduce the size of the

data set and make relationships more difficult to discern.

### 5.2.2 Limitations inherent to each of the three considered biogeographic indexes

Index $JC$ measures the true similarity between two sets of fossil occurrences and is less reliable if the real biodiversity is unknown, which is likely the case with paleontological data. Indeed, the validity of index $JC$ in paleobiogeographic studies is questionable (Simpson, 1960). The use of the union between two sets of fossil occurrences is problematic when one set is

much larger than the other, and incidentally the SCB is one of the largest biota sets for Early Permian brachiopods, with nearly 150 genera present on the SCB, 3-7 times larger than the average number of genera per plate that the SCB is compared to (Fig.





2b). This effect is apparent in Figure 3 in which $JC \approx 0$ and $\ln(JC) \approx -5$ for many plates. In contrast, the absence of plates with $\ln(SC) \approx -5$ or $\ln(cME) \approx -5$ indicates that the lowest index values for $SC$ and $cME$ are equal to 0, rather than close to 0 (Fig. 3). Due to these limitations, index $JC$ is expected to perform poorly relative to the other indexes (e.g. Fig. 5). Index

$SC$ was developed to extrapolate the true similarity by assuming biodiversity is constant across regions (Simpson, 1960) and thus the calculation is limited by the smaller region. This risks overestimating the true similarity if there are real differences in biodiversity and is susceptible to large variations in value from small variations in set intersection when the smaller biota set is small. Index $cME$ is limited by both of the issues that exist for indexes $JC$ and $SC$, however, averaging each endemism ratio reduces the impact of each limitation. The Raup-Crick index (Raup and Crick, 1979), which considers how widespread

each genus is across a given faunal assemblage and compares similarity between observed sets to similarity between randomly generated sets, could be used in future work as it has shown promise (Schmachtenberg, 2008). Each of the three indexes considered in this study are subject to real-world complications to the relationship they measure, typically due to factors such as land barriers, which lead migration distances to much greater than measured physical distances.

### 5.2.3  Possible interdependence of the considered datasets

While the Early Permian location of the SCB was made independently of faunal data in reconstructions W13, M16 and Y19, all three reconstructions used faunal data in the constraint of other tectonic blocks. Faunal data were used to constrain the location of the Australian block by interpreting paleoenvironments in reconstruction W13 (Wright et al., 2013), so that results may not be entirely independent for that plate. In contrast, faunal affinity was only used to determine connected tectonic plates during Devonian times in reconstruction M16 (Matthews et al., 2016), and to determine that Armorica and Perunica were separate

micro-continents at the Silurian-Devonian boundary in reconstruction Y19 (Young et al., 2019). We note that the dispersal of brachiopod larvae depends on ocean currents (Lam et al., 2018), which would be different based on the location of the SCB and other tectonic blocks in each of the considered tectonic reconstructions (Fig. 1).

The method used to constrain the location of the SCB in M16 is also problematic, as it assumes that Large Low Shear Velocity Provinces (basal mantle structures) have remained stationary since the emplacement of the Emeishan LIP (Domeier

and Torsvik, 2014), which is unnecessary since mantle flow models that predict mobile basal mantle structures are consistent with the record of large volcanic eruptions (Flament et al., 2022).

### 5.3  Comparison to previous work and possible future directions

Similar quantitative methods have been used to assess relationships between biogeographic indexes and distance (Lees et al., 2002; Belasky et al., 2002; Shi, 1993; Shen et al., 2013; Angiolini, 2001), although on smaller spatial scales. Improved data

availability and analytical tools have made these quantitative methods viable for application at the global scale. Here, the SCB was compared to all other tectonic plates, rather than to a selection of geographically close plates. Comparing to a small number of plates makes it possible to define a relative position within a region, whereas the comparison to all other plates makes it possible to elect a preferred location relative to global geography.



The framework developed here could be used to systematically evaluate the compatibility of tectonic reconstructions with
faunal data, or applied to other fields of study such as paleo-oceanography. The framework is highly adaptable and can be
applied to any reconstruction, geological period, and faunal data set that provides location data. Paleobiogeography has been
used successfully as a qualitative method to track past ocean currents (Harper et al., 2005), with this framework providing
the opportunity to extend these past works to larger scale, quantitative studies. Testing tectonic plate locations could also go
beyond testing single tectonic plates, as was done here, by testing faunal affinity for all tectonic plates against all other tectonic
plates across geological times. The framework could also be extended to any suitable fossil type. In principle, it should be
possible to design a framework in which distances between all tectonic blocks would be a function of faunal similarity, which
is one of the observations that led to the establishment of tectonic reconstructions (Jell, 1974). In such a framework, faunal
constraints would be considered alongside geological and paleomagnetic constraints.

## 6  Conclusions

We assessed the Early Permian longitude of the SCB by using a new framework to evaluate the global relationship between
faunal similarity and tectonic distance. We determined that Early Permian longitude of the SCB in the reconstruction of Young
et al. (2019) is more consistent with brachiopod distribution than the Early Permian longitude of the SCB in the reconstruction
of Wright et al. (2013) and Matthews et al. (2016). It was found that the strength of correlation between faunal similarity and
physical distance between the SCB and circum-tethyan tectonic blocks was greatest for Y19. This implies that the SCB could
have been positioned centrally within the Paleo-Tethys Ocean at 103ºE, which is 20-30º further west than the other considered
reconstructions which tend to place the SCB on the margin of the ocean. This likely has important implications for Paleo-Tethys
Ocean circulation and on reconstructions of paleoclimate change throughout the Early Permian. The presented framework is
also a valuable contribution to future studies on paleogeography as it can be extended to refine global tectonic plate locations
based on faunal data.

*Code and data availability.* All data sets and code used for analysis are openly available from Zenodo (Marks and Flament, 2025) as a
collection of Jupyter notebooks that can be used to reproduce the analysis carried out in this contribution, or adapted to perform analysis for
other plate tectonic reconstructions, times, or fossil types. Tectonic reconstruction data used within this manuscript are openly available for
Y19, M16 and W13 at Young et al. (2019), Matthews et al. (2016) and Wright et al. (2013), respectively.

*Author contributions.* RM: Conceptualisation, Methodology, Software, Investigation, Formal analysis, Writing - Original Draft, Writing -
Review and Editing, Visualisation; NF: Conceptualisation, Methodology, Software, Writing - Review and Editing, Visualisation, Supervision;
SL: Conceptualisation, Methodology, Writing - Review and Editing, Supervision; GRS: Conceptualisation, Writing - Review and Editing



*Competing interests.* The authors declare that they have no conflict of interest.



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
