# Peer review of "Early Permian longitudinal position of the South China Block from brachiopod paleobiogeography"

_EGUsphere, 2025_

## Author Response (AR1)

**Point-by-point response to reviewer comments**

We thank the two reviewers for their constructive comments and suggestions, which have been valuable in improving our manuscript. Here we present a point-by-point response to all reviewer comments (reviewer comments in bold, response in plain text) and note corresponding changes to the manuscript (referenced by line numbers in the revised manuscript file).

**Response to RC1:**

Reviewer: This study presents a novel and ambitious approach to constrain the Early Permian longitudinal position of the South China Block (SCB) using brachiopod paleobiogeography, offering a creative solution to the longstanding challenge of reconstructing paleolongitude in deep-time tectonic models. By integrating quantitative faunal similarity indices (Jaccard, Simpson, and cME) with global plate reconstructions, the authors provide a framework that bridges paleobiology and geodynamics, marking a significant methodological advance. The conclusion that the SCB occupied a central position within the Paleo-Tethys Ocean (as per Young et al., 2019) challenges previous marginal placements and has implications for paleoceanographic and climatic interpretations. The open accessibility of the analytical framework further enhances its utility for future studies.

Reviewer: First, the reliance on brachiopod distribution assumes that faunal similarity inversely correlates with physical distance, yet environmental heterogeneity, larval dispersal barriers (e.g., landmasses, currents), and sampling biases (e.g., uneven fossil preservation/collection) could decouple this relationship. While the authors acknowledge these issues, the extent to which they influence the indices—particularly given the SCB's disproportionately large dataset—remains unclear. For instance, the Jaccard index's poor performance highlights the vulnerability of binary presence-absence metrics to sampling disparities, suggesting that results may overemphasize reconstruction Y19's plausibility.

Response: We further discuss these limitations and how they impact faunal similarity correlations with the South China Block (SCB) in the revised manuscript. Some of these concerns are alleviated by using distance limits, which can exclude plates for which decoupling factors such as larval dispersal barriers are a major concern. Additionally, we now use a Spearman rank correlation which further reduces impact of these decoupling factors as it is more robust to outliers than the Pearson correlation we previously used (discussed in greater depth below). It is unclear to us how these known

limitations emphasise the plausibility of any of the three considered reconstructions. Changes related to this comment can be found:

- On lines 406-414, for limitations of index *JC* based on sample size discrepancies
- On lines 422-432, for discussion on larval dispersal barriers decoupling the relationship
- On lines 439-442, for differing ocean current scenarios

Reviewer: Second, the tectonic models themselves inherit uncertainties. The assumption of fixed LLSVPs in Matthews et al. (2016) versus their potential mobility in Young et al. (2019) reflects debated geodynamic hypotheses, yet the study does not fully disentangle how these contrasting assumptions propagate into the faunal-distance correlations.

Response: The assumptions/uncertainties within the tectonic models are part of determining tectonic plate locations. The central idea of the paper is to use faunal similarity-distance correlations to test these locations. We clarified this at the end of the introduction (lines 51-58).

Reviewer: Additionally, the choice of 277 Ma as a representative time slice overlooks temporal dynamics within the ~27 Myr Early Permian, during which climatic shifts (e.g., deglaciation) and biotic turnover could skew biogeographic patterns.

Response: We agree and have split the data in the revised manuscript, performing the analysis for two distinct periods: Asselian-Sakmarian times (cooler climate) and Artinskian-Kungurian times (warmer climate). We focus the revised manuscript on the Artinskian-Kungurian times for which more data are available and reconstruction W13 is explicitly, as opposed to interpolated for Asselian-Sakmarian times. This change demonstrates the adaptability of the framework. There are major revisions to the paper because of this change (although the main results remain the same), and the most significant changes can be found:

- On lines 99-111, justification for the two time periods
- All figures from the original manuscript have been changed
- Figure 7 has been added to display a subset of results for Asselian-Sakmarian times with corresponding results text on lines 259-268 and discussion on lines 316-346.

Reviewer: A critical but unaddressed issue lies in the taxonomic accuracy of brachiopod genera extracted from the Paleobiology Database. Fossil identifications in large-scale databases are prone to errors due to misclassification, synonymies, or outdated taxonomy. For example, brachiopod genera with overlapping morphological features or poorly preserved specimens

may be mis-assigned, directly distorting faunal similarity calculations. Such inaccuracies could artificially inflate or diminish correlations between biogeographic indices and physical distance. To strengthen the robustness of the analysis, future iterations of this framework should involve systematic reevaluation of the brachiopod taxonomic data by domain experts to resolve ambiguities and validate species assignments. I believe some of the authors are brachiopod experts, not sure if they reviewed the taxonomy of the genera extracted from PBDB.

Response: We clarify in the revision that the Early Permian brachiopod records from the PBDB are expected to be reliable as they are based solely on published taxonomies. We use genus-level records rather than species-level (higher taxonomic rank is less susceptible to taxonomic biases), and we exclude any uncertain genera. Finally, we perform the analysis at the family level (taxonomically, one rank above genus) as well, with the coarser taxonomic resolution having less uncertainty in identification than genus. The results obtained for this analysis showed family-level similarity corroborated the results for genus-level similarity, and this analysis again demonstrates the flexibility of the framework we introduce in the contribution. Changes related to this comment can be found:

- On lines 127-133 for clarification on taxonomic reliability
- On line 136 for the exclusion of uncertain genera in the data download
- Figure 8 has been added as a discussion figure showing the family-level analysis, this is a sensitivity analysis to assess the robustness of the framework to changing taxonomic resolution. Text related to Figure 8 can be found on lines 372-382.

Reviewer: Lastly, the statistical approach—while rigorous—simplifies complex biogeographic processes into linear relationships. Nonlinear effects (e.g., threshold distances for provinciality) or geographic barriers (e.g., continental shelves) may distort correlations, particularly for marine taxa like brachiopods.

Response: The relationship between faunal similarity and physical distance is complex and may not be strictly linear or logarithmic, however, it is still expected to be monotonically decreasing. To account for the complexity of the relationship and test only for a monotonic relationship, we have changed the statistical testing to be based on the Spearman rank correlation. Explanation for this choice in statistical test can be found on lines 196-208.

In addition to this, we include figures here (Reply Figure 1) presenting a comparison of the Spearman correlation results to Pearson correlation results on both the linear space data and log transformed data. We found that the difference between the three statistical methods are often quite minor, with  $r_{\rm s}$  varying by

Reply Figure 1: Comparison of the strength of correlation using Spearman rank correlation, linear Pearson correlation, and Pearson correlation of natural logarithm transformed data between faunal similarity and physical distance from the SCB for reconstruction Y19 during the Artinskian-Kungurian at all distance limits.

Reviewer: The framework's scalability to other taxa/periods, though promising, requires validation against independent datasets (e.g., paleomagnetic or stratigraphic constraints).

Response: We agree that the framework is best used to provide evidence supporting plate tectonic configurations using faunal data in conjunction with other, independent datasets. We clarify this on lines 474-476.

**Response to RC2:**

Reviewer: This interesting and innovative manuscript studies the palaeolongitude of the South China Block (SCB) during the Early Permian by investigating the faunal affinity of brachiopods between the SCB and other tectonic plates. Based on three different paleogeographic reconstructions, the manuscript employs strict statistical analysis to examine the relationship between brachiopod faunal similarities and physical distances. The study supports that the SCB were positioned in the central part of the Palaeo-Tethys Ocean, rather than at its periphery, challenging the conventional views. However, some weaknesses remain in the research methods.

Reviewer: Comparing faunal affinities between SCB and other plates across the entire Early Permian (spanning ~17 Ma) is problematic. The SCB remained in the palaeoequatorial region throughout this interval, maintaining consistent Tethyan warm-water brachiopods. In contrast, other tectonic units, particularly the Cimmerian Terranes, underwent significant faunal transitions, evolving from Gondwanan cold-water taxa to cool- or even warm-water elements throughout the early Permian. Thus, the brachiopod faunas of these mobile blocks could shift from being very different to closer to those of the SCB over this timespan. To obtain more reliable results, I strongly recommend dividing the early Permian into two intervals (Asselian-Sakmarian and Artinskian-Kungurian) for separate analyses.

Response: We agree and have split the data in the revised manuscript, performing the analysis for two distinct periods: Asselian-Sakmarian times (cooler climate) and Artinskian-Kungurian times (warmer climate). Further discussion of this and related changes can be found above.

We also added further discussion on the uncertain position of the Cimmerian Terranes, particularly relating to other reconstructions (not included in this study) that use faunal affinity to define a position different to those seen in the three reconstructions we consider. This discussion can be found on lines 455-464.

Reviewer: As noted by the authors, the North American brachiopod faunas exhibit significant diversity during the Early Permian. However, they were excluded from the analyses due to their far distance (>12,000 km) from the South China Block in

all three reconstruction maps. The North America Plate was situated in the palaeoequatorial region, and its faunas likely maintained biogeographic connections with South China via ocean currents. Thus, its inclusion in the analyses would provide a more comprehensive assessment of faunal affinities versus distances.

Response: For the North American plate, there is a discrepancy between the physical distance between the SCB and North America, and the distance of dispersal pathways for marine fauna. The North American brachiopod faunas are on the Panthalassan coastline and could disperse either across the Panthalassa Ocean or north around the Siberian block (Fig. 1 in the manuscript), both of which are quite large distances. Due to these barriers to migration, the genus-level faunal similarity has been found to be considerably limited (Waterhouse and Bonham-Carter 1975), as expected by the great distances. However, the physical distance between these two plates, as determined using pyGPlates, is directly across the Tethys. This makes the physical distance relatively short when compared to the distance travelled for dispersal between the two regions. This justification is clarified on lines 424-432.

Reviewer: Another issue concerns the inconsistent distance thresholds applied in the faunal similarity analyses. In W13 (Fig. 3), the data appear to have a global scope, with distances extending up to 20,000 km. In contrast, Y19 (Fig. 6) restricts the analysis to plates within an 8,000 km distance limit. In addition, relationships between biogeographical indexes and distance based on M16 are absent. What is the basis for the choice of distance limits in these analyses?

Response: We clarify and extend our implementation of distance limits in the revised manuscript. Figure 3 illustrates that a global analysis does not necessarily provide a valuable correlation as it is impacted by various issues, such as the abovementioned differences between physical distances and dispersal pathways. For all three reconstructions, we examine the relationship between biogeographical indexes at all possible distance limits, with the lower limit being the closest plate to the SCB and upper limit being the global plate distribution. Figure 4 illustrates this, with distance limits on the X-axis, and correlation coefficient between biogeographic indexes and physical distance for these distance limits on the Y-axis. While we present the correlation for Y19 with a 10,000 km distance limit in Figure 6, this is what we have determined as the most successful reconstruction and distance limit within this framework. To put these results in context, Figure 4 shows the strength of the correlation for all considered reconstructions and distance limits. The clarification and extension of distance limits are discussed:

- On lines 183-192 for the new distance limits
- Figure 4 caption

- Vertical orange lines in Figure 4 panels now highlight that the correlation shown in Figure 6 is a snapshot of results that are contained within Figure 4.

Reviewer: The authors consider that the latitudinal positions of the SCB were relatively stable in three configurations, its longitudinal variation significantly affects distance-based analyses. However, the latitudinal uncertainties of the SCB affect the distance of other plates to its north and south. For example, the Australian Plate exhibits substantial discrepancies in distance between the SCB in different reconstructions: its centroid ranges between 4000-6000 km in W13, but 6000-8000 km in Y19 and M16. Considering the high diversity of brachiopods of the Australian Plate, the differences in the distance could have a large impact on the results.

Response: We agree that the SCB is largely in the Southern Hemisphere in reconstruction W13, which affects the distance between the SCB and the Australian plate (as well as other plates). We clarify this and emphasise that while the comparison is not a strict comparison of paleolongitude between reconstruction W13 and reconstructions M16 and Y19, it is still a valuable comparison of suitability within the global plate tectonic configuration. Additionally, while there is latitudinal variation between the reconstructions, the SCB remained as a distinct biogeographic province within the paleoequatorial biogeographic realm. This can be found on lines 310-315.

Reviewer: In addition, the faunal appearances of Western Australia and eastern Australia are really different, it is unclear whether this study treats the Australian Plate as a single plate or divides it into two geographic units.

Response: We considered the Australian Plate as a single plate. We acknowledged in the discussion (section 5.2.1 Faunal Provinciality) that this is not ideal for large plates that likely contain multiple, distinct faunal provinces as is the case for the Australian plate. Breaking the analysis down to faunal provinces could be done in future work. We consider this to represent future work and have made no changes to the revised manuscript in relation to this comment.

Reviewer: For Fig. 1, I suggest to add the citations and abbreviations in the blank space of each map, such as Young et al., (2019, Y19), which will make the article more readable.

Response: We agree that this change improves readability and have updated Figure 1 accordingly.

Reviewer: For Fig. 2a, I am wondering if the number of plates includes all plates at that range or only those containing brachiopods. Displaying the number of plates with brachiopod records would be more meaningful, as only those would be used in the analysis.

Response: Figure 2 only considers plates which contain brachiopod occurrences and the caption for Figure 2 has been updated to clarify this.